# Performance of a Full-Scale Vermifilter for Sewage Treatment in Removing Organic Matter, Nutrients, and Antibiotic-Resistant Bacteria

Victor Gutiérrez [1], Naomi Monsalves [1,2], Gloria Gómez [1,2] and Gladys Vidal [1,2,*]

[1] Environmental Engineering & Biotechnology Group (GIBA-UDEC), Environmental Science Faculty, Universidad de Concepción, Concepción 4030000, Chile

[2] Water Research Center for Agriculture and Mining (CRHIAM), ANID Fondap Center, Victoria 1295, Concepción 4030000, Chile

* Correspondence: glvidal@udec.cl

**Abstract:** The vermifilter (VF) is regarded as a sustainable solution for treating rural sewage. However, few studies have investigated the performance of a full-scale vermifilter. The objective of this study is to evaluate the performance of a full-scale vermifilter in reducing organic matter, nutrients, and antibiotic-resistant bacteria contained in sewage. Influent and effluents were obtained from a rural sewage treatment plant using a VF and UV disinfection system. The results show a significant removal ($p < 0.05$) of chemical organic demand (COD) (77%), biochemical oxygen demand ($BOD_5$) (84%), total nitrogen (TN) (53%), and total phosphorus (36%). Seasonality is an influential variable for COD, $BOD_5$, and TN removal. In addition, the molecular weight distribution shows that the VF does not generate a considerable change in the distribution of organic matter (COD and total organic carbon (TOC)) and $NH_4^+$-N. The UV disinfection system eliminated 99% of coliform bacteria; however, they are not eliminated to safe concentrations. Therefore, it is possible to detect bacteria resistant to the antibiotics ciprofloxacin, amoxicillin, and ceftriaxone at 63.5%, 87.3%, and 63.5%, respectively, which were detected in the effluents. This study shows the potential of a system for the removal of pollution and the need to optimize the VF to be a safe treatment.

**Keywords:** sewage treatment; full-scale vermifilter; antibiotic-resistant bacteria; molecular weight distribution

## 1. Introduction

In 2010, the World Health Organization (WHO) recognized access to drinking water and sanitation as a human right. However, in 2020, 45% of the sewage generated in the world was still discharged without safe treatment [1]. It has also been observed that sanitation coverage is often much lower in rural areas than in urban areas. The lack of adequate sanitation services can lead to environmental pollution, causing water quality deterioration, biodiversity loss, and changes in an ecosystem's structure and function [2]. In addition, because sewage contains a great variety of fecal microorganisms and pathogens, the creation of an unhealthy environment leads to the transmission of diarrheal diseases such as cholera, dysentery, and typhoid fever [1,3]. Meanwhile, a lack of sewage treatment contributes to the propagation of antibiotic resistance.

The emergence of antibiotic-resistant bacteria (ARB) and their dissemination through the environment—recognized as one of the main problems of the 21st century [4]—limit the treatment of infectious diseases and increase the probabilities of morbidity and mortality in the population [5]. ARB can survive and multiply in the presence of antibiotics thanks to antibiotic resistance genes (ARG), which encode proteins that participate in various resistance mechanisms and can be transferred to other bacteria to make them resistant [6].

Therefore, the search for decentralized technologies that can be applied in rural areas with low populations and scattered homes to minimize the risks of inefficiently treated

sewage discharge has become a priority [7–10]. The technologies that comply with this requirement include constructed wetlands (CW), upflow anaerobic sludge blankets (UASB), and moving-bed biofilm reactors (MBBR). However, they may have drawbacks that increase their operating costs, such as clogging of the substrate in CW [11], sludge flotation and biomass washing at UASB [12], and the need for aireation in MBBR [13]. On the other hand, the vermifilters (VFs) function as aerobic biofilters due to the burrowing behavior of the earthworms that increase oxygen. Therefore, they do not require induced aeration or pumping systems that increase the cost of treatment [14]. Furthermore, the earthworms are able to eat all the suspended particles screened on the filter bed and avoid the generation of sludge from a conventional biofilter [15]. The VF is a bio-oxidative process based on the symbiotic relationship between earthworms and microorganisms used to biochemically degrade waste materials [16]. Because of this, VFs can be a treatment that is competitive with rural wastewater treatment plants (WWTP) due to their cost savings and ecological characteristics [15,17]. Muga and Mihelcic [18] deem VFs as sustainable technologies, as they maintain economic and environmental welfare as well as seek equitable social progress.

High VF contaminant removal performance has been observed with chemical organic demand (COD), nitrogen, and phosphorous removal rates of 75–90%, 32–85%, and 39–95%, respectively [19–21]. They have also proved to be efficient in the removal of total coliform (CT), fecal coliform (CF), and ARB, with rates of 58–99%, 52–99%, and 100%, respectively [9,22,23]. However, there have been few studies on full-scale VFs, as most are conducted at a laboratory scale and operated on a short-term basis under controlled experimental conditions [24,25]. Furthermore, there are no studies associated with joint coliform and ARB removal efficiencies.

Therefore, the objective of this study is to assess the organic matter, nutrients, and ARB removal efficiency of a full-scale VF that treats sewage. In addition, this study will evaluate the performance and influence of seasonality through molecular weight distribution analysis with respect to organic matter and nutrient removal efficiency.

## 2. Materials and Methods

### 2.1. Design and Operating Conditions

This study was carried out in a rural WWTP in Copiulemu Commune, Concepción, Biobío Region, Chile. This WWTP used a solid removal pretreatment system, a vermifilter (VF), and a UV radiation system as a tertiary treatment. The VF is a vertical subsurface flow system, the main characteristics of which are summarized in Table 1. Sewage enters a rotary filter with a helical plate to carry solids to a container outside the VF. The sewage is stored in a holding tank and transferred by two pumps to the upper part of the VF, distributing the sewage over the surface via 24 sprinklers.

**Table 1.** Summary of vermifilter design and operating parameters.

| Parameter | Unit | Value |
|---|---|---|
| Vermifilter volume | $m^3$ | 442 |
| Vermifilter area | $m^2$ | 276 |
| Vermifilter height | m | 1.60 |
| Influent flow | $m^3/d$ | 160 |
| HLR | $m^3/m^2d$ | 0.60 |
| OLR | kg COD/$m^2$d | 0.50 |
| Carbon/Nitrogen ratio | - | 9 |
| Temperature | °C | 21 |
| Active layer | - | woodchip |
| Active layer height | m | $0.90 \pm 0.12$ |
| Species | - | *Eisenia foetida* |
| Earthworm density zone A | worm/$m^3$ | ~1100 |
| Earthworm density zone B | worm/$m^3$ | ~7000 |

Note: HLR: hydraulic loading rate; OLR: organic loading rate; COD: chemical oxygen demand.

The VF uses a flow rate, hydraulic loading rate (HLR), and organic loading rate (OLR) of 160 m$^3$/d, 0.6 m$^3$/m$^2$d, and 0.5 kg COD/m$^2$d, respectively. Figure 1 shows a schematic representation of the active layer, which included the filter material with a lower stone layer and upper woodchip layer (active layer) of 0.92 ± 0.12 m. The active layer contains the *Eisenia foetida* species, which are distributed between different zones. The upper zone (zona A) has a height of 104 cm of woodchips and contains~1100 earthworms per cubic meter, and the lower zone (zone B) has a height of 80 cm of woodchips and contains~7000 earthworms per cubic meter. In addition, the disinfection system operates with a gravitational flow of 4.18 m$^3$/h. It is composed of 15 low-pressure and high-intensity mercury lamps, with an input power of 87 W and an output power of 28 W.

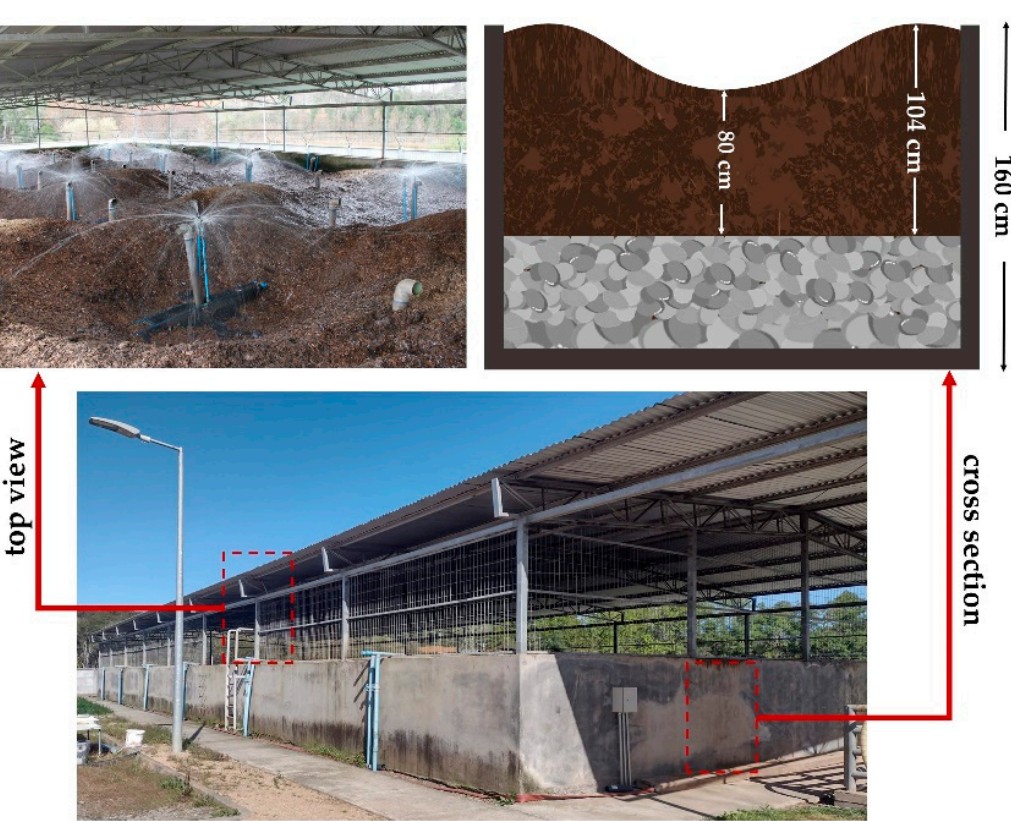

**Figure 1.** Schematic representation of the active layer.

### 2.2. Monitoring Strategy

The monitoring strategy involved taking samples between May 2022 and January 2023, covering the fall, winter, spring, and summer seasons of the Southern Hemisphere. Figure 2 shows a schematic diagram of the monitoring strategy, which included the collection of samples at the pretreatment outlet (IN), VF outlet (E1), and UV outlet (E2). Meanwhile, the solid samples were obtained by inserting a tube (12 cm in diameter and 100 cm in length) into the active layer. Woodchips were collected from the inside of the tube to calculate earthworm density. All the samples were stored for less than 24 h at 4 °C prior to their analysis.

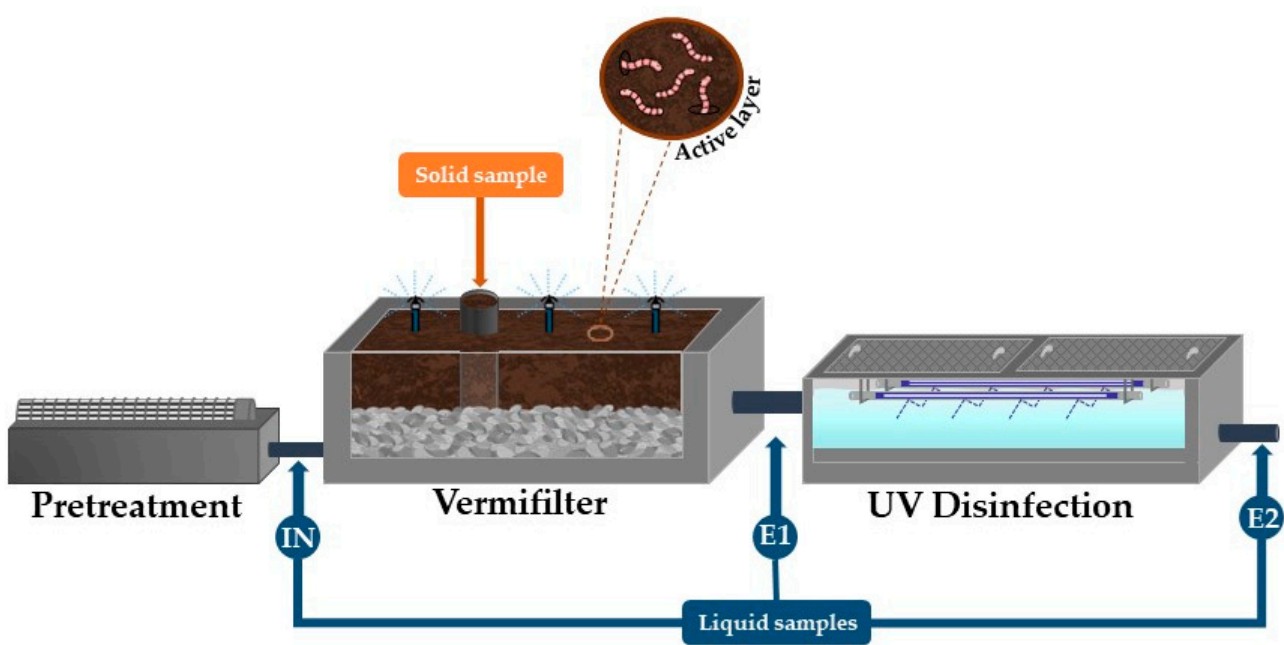

**Figure 2.** Schematic representation of the monitoring strategy. IN: pretreatment outlet; E1: vermifilter outlet; E2: UV (ultraviolet) outlet.

*2.3. Physicochemical Analysis*

2.3.1. Liquid Samples

To determine the quality parameters of IN, E1, and E2, the samples were filtered with a Whatman membrane with a pore size of 0.7 μm and were analyzed based on the protocol described in the standard method [26]. The in situ parameters of pH, temperature (T), dissolved oxygen (DO), electrical conductivity (EC), and oxidation-reduction potential (ORP) were measured using a multiparametric OAKTON-PC650 (Eutech Instruments; Singapore). DO was measured using an Oxi 330i handheld oximeter (WTW, Xylem Analytics Germany Sales, Oberbayern, Germany). The organic matter present in the samples was determined in the form of COD (colorimetric method, 5220-D), total organic carbon (TOC) (catalytic combustion oxidation and NDIR detection, TOC analyzer-LCPH, Shimadzu, Kyoto, Japan), and biological oxygen demand ($BOD_5$) (azide-modified Winkler method, 5210-B). Solids were measured based on total suspended solids (TSS) and volatile suspended solids (VSS). TSS was determined using gravimetric methods in which the sample was filtered (1.5 μm) and dried at 105 °C. To determine VSS, the samples were dried for 30 min at 550 °C and subtracted from the TSS value (gravimetric method, 2540-D). The analyzed nutrients were measured in the form of $NH_4^+$-N, $NO_3^-$-N, $NO_2^-$-N, $PO_4^{3-}$-P (Shimadzu UV 1800 UV–Vis spectrophotometer, Kyoto Japan), total nitrogen (TN) (Spectroquant-Nova 60, Merck kits, Darmstadt, Germany), and total phosphorous (TP) (Spectroquant-Nova, Merck kits, Darmstadt, Germany).

2.3.2. Solid Samples

The pH, EC, and ORP values were measured by diluting the active layer in distilled water at a ratio of 1:10 and stirring at 200 rpm for 1 h [8]. Moisture content was determined by the weight loss of the sample at 105 °C in 24 h [27]. The biofilm analyses were conducted by extracting 20 g of woodchips, which were then suspended in 20 mL of distilled water and sonicated for 20 min to quantify them as volatile solids (VS) per area [26,28].

### 2.4. Molecular Weight Distribution Analysis of Liquid Samples

The IN and E2 samples underwent COD, TOC, and $NH_4^+$-N molecular weight distribution analysis. To this end, ultrafiltration (UF) was performed in a 450-mL shaken cell (Advantec UHP 76) at 20 °C using nitrogen gas. The UF was carried out using three cellulose membranes with nominal molecular weight cut-offs of 10,000 Da, 5000 Da, and 1000 Da, allowing four different fractions to be obtained: compounds greater than 10,000 Da ("10,000 Da"), compounds between 5000 and 10,000 Da ("5000–10,000 Da"), compounds between 1000 and 5000 Da ("1000–5000 Da"), and compounds below 1000 Da ("<1000 Da") [29,30]. Each of the obtained fractions underwent COT, COD, and $NH_4^+$-N quantification, as mentioned in Section 2.3.1.

### 2.5. Microbiological Analysis of Liquid Samples

The FC, TC, and ARB content of the IN, E1, and E2 samples was analyzed in accordance with the protocol described in the standard method [26]. FC and TC were determined by means of the multiple tube technique using the most probable number methodology (MPN/100 mL). The presence of bacterial groups was determined by means of a presumptive and confirmatory test (Standard Method 9221-TC). ARB concentrations, meanwhile, were determined by antimicrobial susceptibility testing (AST) using the plate count technique [31]. MacCONKEY agar (Merck, Darmstadt, Germany) was used as a culture medium and the plates were supplemented with the antibiotics of amoxicillin (AMX), ceftriaxone (CTX), and ciprofloxacin (CIP) at concentrations of 32 µg/mL, 4 µg/mL, and 2 µg/mL, respectively [32]. Dilutions were prepared for IN, E1, and E2 samples and were incubated at 30 °C for 24 h. After incubation, the colonies capable of resisting the action of the antibiotics were counted, so this technique expresses the ARB concentration in colony-forming units (CFU/100 mL).

### 2.6. Statistical Analysis

Statistical analyses of the organic matter, nutrient, ARB, and coliform removals as well as the effect of seasonality were performed using RStudio version 1.2.1335, with a significance level of $p = 0.05$. Shapiro–Wilk and Fligner–Killen tests were conducted to analyze the normality and homogeneity of variance, respectively. Next, an ANOVA test was performed for the data with a normal distribution and this was followed by a Kruskal–Wallis test for data without a normal distribution.

## 3. Results and Discussion

### 3.1. Evaluation of Vermifilter Performance in Organic Matter and Nutrient Removal

Table 2 shows the results regarding the physicochemical and microbiological parameters in IN, E1, and E2. During the 6 months of monitoring, the full-scale VF presented statistically significant differences ($p < 0.05$) between IN and E2 in pH decrease ($7.8 \pm 0.4$ to $6.8 \pm 0.3$), ORP increase ($-25.4 \pm 141.7$ to $245.9 \pm 127.5$), and turbidity decrease (68.6%). Similarly, statistically significant removal efficiencies ($p < 0.05$) between IN and E2 were found for $NH_4^+$-N (81%), TN (53%), $PO_4^{3-}$-P (34%), TP (36%), COD (77%), $BOD_5$ (84%), TSS (78%), and VSS (79%). The high standard deviation values of ORP, $BOD_5$, and TSS, can be explained by variations in the influent, which affect the storage levels and the sedimentation of solids before being transferred by two pumps to the VF [33].

**Table 2.** Physicochemical characterization of IN, E1, and E2 during the monitoring period.

| Parameter | | Unit | IN (Mean $\pm$ SD) | Effluent (Mean $\pm$ SD) | |
|---|---|---|---|---|---|
| | | | | E1 | E2 |
| In situ | T | °C | $17.1 \pm 3.7$ | $15.9 \pm 3.6$ | $15.8 \pm 3.4$ |
| | pH | - | $7.8 \pm 0.4$ | $6.9 \pm 0.2$ | $6.8 * \pm 0.3$ |
| | ORP | mV | $-25.4 \pm 141.7$ | $181.8 \pm 77.3$ | $245.9 * \pm 127.5$ |
| | EC | $\mu$S/cm | $1.6 \pm 0.2$ | $1.0 \pm 0.4$ | $1.2 \pm 0.1$ |
| | DO | mg/L | $1.2 \pm 0.5$ | $5.1 \pm 0.6$ | $5.0 * \pm 0.8$ |
| | Turbidity | NTU | $210.8 \pm 59.4$ | $61.2 \pm 22.6$ | $66.2 * \pm 30.3$ |
| Nutrients | $NH_4^+$-N | mg/L | $104.9 \pm 35.3$ | $23.0 \pm 10.7$ | $19.8 * \pm 8.7$ |
| | TN | mg/L | $114.4 \pm 32.5$ | $57.5 \pm 8.1$ | $53.7 * \pm 6.0$ |
| | $PO_4^{3-}$-P | mg/L | $10.3 \pm 2.6$ | $6.5 \pm 2.4$ | $6.8 * \pm 1.0$ |
| | TP | mg/L | $12.5 \pm 2.0$ | $7.9 \pm 2.9$ | $8.0 * \pm 1.5$ |
| Organic matter | COD | mg/L | $885.8 \pm 89.9$ | $286.0 \pm 60.7$ | $204.2 * \pm 38.4$ |
| | $BOD_5$ | mg/L | $486.3 \pm 123.1$ | $109.6 \pm 27.5$ | $79.4 * \pm 32.6$ |
| | TSS | mg/L | $239.1 \pm 101.4$ | $67.3 \pm 16.5$ | $53.4 * \pm 17.9$ |
| | VSS | mg/L | $186.6 \pm 68.1$ | $50.8 \pm 18.9$ | $39.5 * \pm 14.1$ |
| Microbiological | TC | $\log_{10}$(MPN/100 mL) | $8.7 \pm 0.7$ | $6.5 \pm 0.9$ | $5.2 * \pm 1.0$ |
| | FC | $\log_{10}$(MPN/100 mL) | $7.8 \pm 1.0$ | $5.9 \pm 1.0$ | $4.4 * \pm 1.3$ |

Note: IN: Influent; E1: VF effluent; E2: UV effluent; SD: Standard deviation; NTU: Nephelometric Turbidity Unit; T: temperature; ORP: oxidation-reduction potential; EC: electrical conductivity; DO: dissolved oxygen; TN: total nitrogen; TP: total phosphorus; COD: chemical oxygen demand; $BOD_5$: biological oxygen demand, at five days; TSS: total suspended solids; VSS: volatile suspended solids; TC: total coliforms; FC: fecal coliforms; MPN: Most Probable Number, * statistically significant differences ($p > 0.05$) between IN and E2.

### 3.1.1. Organic Matter

The VF achieved statistically significant removal efficiencies ($p < 0.05$) for COD and $BOD_5$ of 77% and 84%, respectively. These results are in line with those reported by authors such as Wang et al. [34], Zhao et al. [35], Tahar et al. [10], and Karla et al. [36], who reported COD and $BOD_5$ removal efficiencies above 75%. Organic matter removal in the VF is determined by the physicochemical modifications of the active layer, burrowing behaviors that promote the development of aerobic microorganisms, and earthworm excreta that modify the microbiome [24,37,38].

Table 3 presents different pilot- and full-scale VFs with their respective operating parameters and organic matter and nutrient removal efficiencies. With respect to active layer material, the woodchip allowed removal efficiencies of over 77% of COD and $BOD_5$. This coincides with results obtained from pilot-scale VFs, which report efficiencies of over 75% [25,35,39]. These results can be explained by how woodchip has been reported to have a hydraulic conductivity of 250–700 mm/h which prevents clogging [40] and has adsorption properties that allow contaminant removal [10,41].

Table S1 shows the physicochemical properties of the active layer at different earthworm densities. The high-density zone (Zone B) presents a pH of $4.8 \pm 0.5$ and an ORP of $367.5 \pm 38.7$ mV, while the zone with a low earthworm density (Zone A) presents a pH of $3.9 \pm 0.2$ and an ORP of $239.2 \pm 29.6$ mV. The increase in ORP suggests that a greater earthworm density increases the oxygen concentration and contributes to the degradation of organic matter. Regarding the pH in the layer that is active, the organic oxidation can lower alkalinity and generate organic acids that lower this pH [42]. However, this process can be neutralized by the calcium content in the earthworm gut [43]. To explain the increase in pH in Zone B, Singh et al. [42] and Hughes et al. [44] show that pH levels below 4.5 limit the performance of a VF. On the other hand, Tahar et al. [10] indicate that high removal efficiencies are achieved at 5000–10,000 worm/m$^3$. Therefore, although densities of 1100 worm/m$^3$ are sufficient to maintain aerobic environments with high oxidation potentials ($239.2 \pm 29.6$ mV), the worms may have problems neutralizing the active layer and this may affect COD and $BOD_5$ removal efficiency.

Regarding the operating parameters, HLR and OLR correspond to influential parameters in the process of organic matter removal [45,46]. In general, Table 3 shows that pilot-scale VFs operate with HLRs less than 1 $m^3/m^2$d and OLRs less than 1 kg COD/$m^2$ [16,27,47]. Ghasemi et al. [45] and Jiang et al. [14] reported that operating with HLRs less than 2.5 $m^3/m^2$d and OLRs up to 0.6 kg COD/$m^2$d allows for the optimal growth of the microbial community without the system becoming clogged with organic matter. On this note, Liu et al. [24], Ghasemi et al. [45], and Pous et al. [39] achieved COD and $BOD_5$ removal efficiencies greater than 70% in systems operating with HLRs over 1.3 $m^3/m^2$d and OLRs over 1 $m^3/m^2$. This is related to the use of systems complementary to the VF such as zooplankton biofilters and recirculation processes.

**Table 3.** Removal of organic matter and nutrients in pilot- and full-scale vermifilters with different operational configurations.

| Scale | Volume ($m^3$) | Active Layer | Density (worm/$m^3$) | HLR ($m^3$/$m^2$d) | OLR (kg COD /$m^2$d) | Organic Matter (% Removal) | | Nutrients (% Removal) | | Ref. |
|---|---|---|---|---|---|---|---|---|---|---|
| | | | | | | COD | $BOD_5$ | TN | TP | |
| Pilot | 0.50 | Woodchips | 13,263 | 1.30 | 1.20 | 91 | 91 | 3 | 15 | [39] |
| Pilot | 1.50 | Compost/soil | 5000–6000 | 2.00 | 0.37 | 60 | - | - | - | [45] |
| Pilot | - | Woodchips | - | 0.48 | 1.70–2.93 | 84 | - | 65 | 70 | [25] |
| Pilot | 0.10 | Sand | 285 | 0.08 | 0.07 | 68 | - | 60 | 45 | [47] |
| Pilot | 0.30 | Organic fraction/vermigratings | 7143 | 1.00 | 0.24 | 78 | 88 | - | - | [27] |
| Pilot | 0.02 | Vermigratings | 30,000 | 1.00 | 0.45 | 74 | 90 | - | - | [11] |
| Pilot | 0.10 | Soil/woodchips | 8929 | 1.05 | 0.11–0.43 | | - | 61 | 62 | [48] |
| Pilot | 0.45 | Soil/woodchips | 3061 | 0.06 | 0.01–0.02 | 81 | - | 80 | 81 | [35] |
| Full-scale | 4.20 | Ceramsite | 16,000 | 4.20 | 0.39 | 68 | 78 | - | - | [24] |
| Full-scale | 433 | Woodchips | A:1100 B: 7000 | 0.50 | 0.60 | 77 | 84 | 53 | 36 | Present study |

Note: T: temperature; HLR: hydraulic loading rate; OLR: organic loading rate; COD: chemical oxygen demand; $BOD_5$: biological oxygen demand, at five days; TN: total nitrogen; TP: total phosphorus; A: zone A; B: zone B.

### 3.1.2. Nutrients

Table 2 shows that the VF achieved TN, $NH_4^+$-N, $PO_4^{3-}$-P, and TP removal efficiencies of 53%, 80%, 34%, and 36%, respectively, which were statistically significant ($p < 0.05$) between IN and E2. These results are in line with the findings of Huang et al. [48], Lavrnic et al. [47], and Tahar et al. [10], who found similar removal efficiencies. The main nitrogen transformation mechanisms occur through nitrification and denitrification [14]. Nitrification in the VF occurs by means of autotrophic microorganisms that develop in the presence of aerobic conditions generated by the burrowing behavior of earthworms [46]. This explains the decrease in $NH_4^+$-N from $104.9 \pm 35.3$ mg/L to $19.8 \pm 8.7$ mg/L and the increase in $NO_3^-$-N from $0.7 \pm 0.6$ mg/L to $29.6 \pm 9.7$ mg/L in E2. In contrast, the decrease in TN from 114.4 mg/L to 53.7 mg/L indicates the presence of denitrification processes, which occur in the guts of the earthworms, as they store denitrifying bacteria [35,46].

Regarding the operating parameters (Table 3), TN removal has also been reported by authors such as Huang et al. [48], Arora and Kazmi et al. [16], and Lavrnic et al. [47] when operating with HLRs between 0.1 $m^3/m^2$d–1.0 $m^3/m^2$d and OLRs between 0.01–0.6 Kg COD/$m^3$d. VFs that operate within these ranges optimize the formation of microorganism colonies in the active layer that participate in nitrification [15,46]. Meanwhile, Thompkins et al. [25] and Ghasemi et al. [45] report TN removal efficiencies greater than 60% at HLRs greater than 1.0 $m^3/m^2$d and OLRs greater than 1 kg COD/$m^2$d, which is attributed to the use of treatment systems complementary to the VF.

Phosphorous removal depends mainly on the adsorption capacity of the active layer [43]. Similar to this VF study, other studies using a pilot scale have achieved high removal efficiencies using woodchips [16,25,35,48]; however, Pous et al. [39] reported removal efficiencies of only 15% in a pilot-scale VF. This result may be associated with earthworm density, as Kumar et al. [22,49] observed TP increases when using a density of 10,000 worm/$m^3$ in a laboratory-scale VF. Earthworms have microorganisms and enzymes capable of mineralizing TP and leaving it available as $PO_4^{3-}$-P. Therefore, earthworm density is considered a fundamental factor among operating parameters.

### 3.1.3. Molecular Weight Distribution Analysis of Organic Matter and Nutrients

Figure 3 shows the performance of the VF in terms of TOC, COD, and $NH_4^+$-N by means of a molecular weight distribution analysis. Organic matter in IN is distributed mainly in fractions between 5000–10,000 Da and >10,000 Da for COD and COT, respectively. Meanwhile, in E2 the COD distribution changes, and this is found mainly in the fractions >10,000 Da. In addition, $NH_4^+$-N is distributed mostly in fractions <1000 Da in IN and E2, at proportions of 60.8% and 55.1%, respectively.

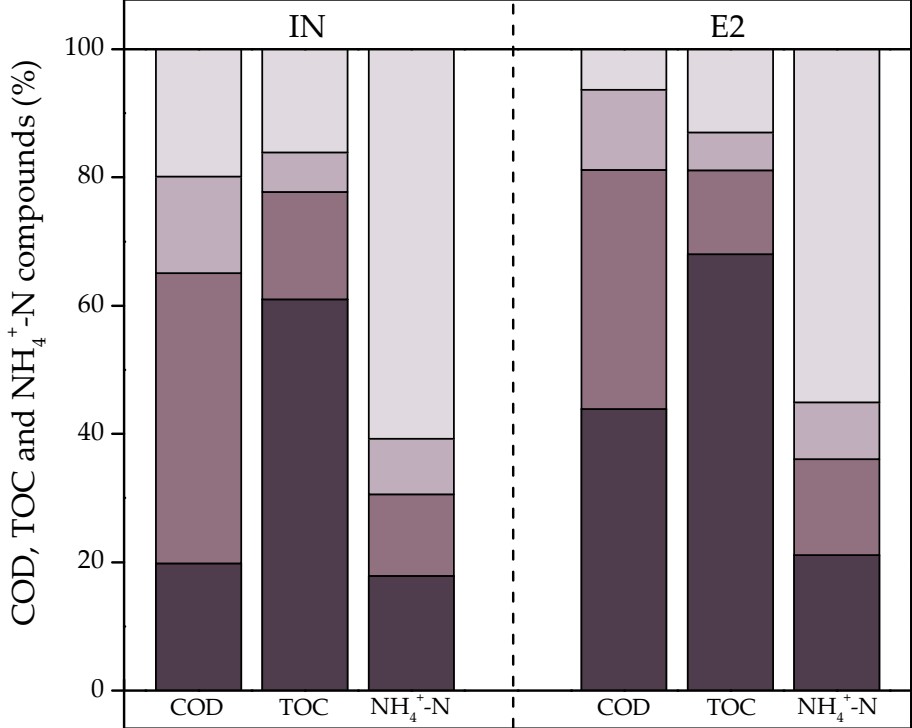

**Figure 3.** Molecular weight distribution of total organic carbon (TOC), chemical oxygen demand (COD), and $NH_4^+$-N in samples of the pretreatment output (IN) and UV output (E2). ▇ = <1000 Da; ▇ = 1000–5000 Da; ▇ = 5000–10,000 Da; ▇ = >10,000 Da.

In E2, there are decreases in COD and TOC in the <10,000 Da fractions of 8% and 4%, respectively. This can be explained by the symbiotic relationship between earthworms and microorganisms that allows the degradation of organic matter in fractions below 10,000 Da to be used as energy and a carbon source [50,51]. In E2, there are also increases in COD and TOC in the >10,000 Da fractions of 24% and 7%, respectively. These results were observed by Wang et al. [51] and Li et al. [52], who reported that organic matter in fractions below 100,000 Da can increase due to biodegradation and formation of aggregates with high molecular weights such as exopolysaccharides and humic substances [51]. However, organic matter with molecular weights above 100,000 Da can be retained in the filter medium. This can explain why the molecular distribution of COD and TOC does not undergo large variations.

With respect to $NH_4^+$-N, the distributions in IN and E2 are similar. In both IN and E2, it is distributed mainly in the <1000 Da fraction. Its distribution in this fraction may be associated with the fact that approximately 90% of the nitrogen comes from urine in the form of $NH_4^+$ [53]. In addition, the digestion process of earthworms leads to the excretion of ammonia and urea, which can be interconverted into $NH_4^+$ depending on the pH [16,54]. In E2, there is a decrease of 5.6% in the <1000 Da fraction, which may be associated with nitrification mechanisms that occur in the system, through which $NH_4^+$-N is converted into $NO_3^-$-N, decreasing its concentration. The fractions 1000–5000 Da and 5000–10,000 Da

remain unchanged. In addition, the increase of 3.3% in the >10,000 Da fraction is associated with the release of proteins and DNA from the biofilm, cell debris, or ARG [51].

Therefore, it can be concluded that the VF does not generate a significant change in the complexity of the organic compounds and $NH_4^+$-N present in sewage. However, it is interesting to note and study the increase in COD and TOC in the >10,000 Da fraction, as it may also be associated with the high presence of TSS in E1 and the low microbiological compound removal efficiency of the UV disinfection system, which will be discussed below.

### 3.2. Effect of Seasonality on the Removal of Organic Matter and Nutrients by a Vermifilter

### 3.2.1. Organic Matter

Figure 4 shows the COD and $BOD_5$ concentrations in IN and E2 in the fall–winter and spring–summer periods, with their respective removal efficiencies. A comparison of the removal efficiencies in each of the evaluated periods shows that the organic matter reduction was statistically significant ($p < 0.05$). In fall–winter, 72% and 78% of COD and $BOD_5$ were eliminated, respectively, while in spring–summer, the respective removals were 81% and 89%. Earthworms are a poikilothermic species; therefore, they are affected by the season's temperatures of the medium [55]. Furlong et al. [56] determined that *Eisenia foetida* lives in an optimum range of 15 °C to 20 °C; thus, temperatures outside this range can affect organic matter, nutrient, and pathogen removal in a VF [16,57]. In the studied full-scale VF, the temperatures were 10 ± 1 °C in fall–winter and 15 °C ± 5 °C in spring–summer. These differences affect the growth and metabolic activity of earthworms and the microorganisms that are part of this symbiotic association [58]. Therefore, below 15 °C it is possible that the decomposition and oxidation processes of the VF will be affected [59].

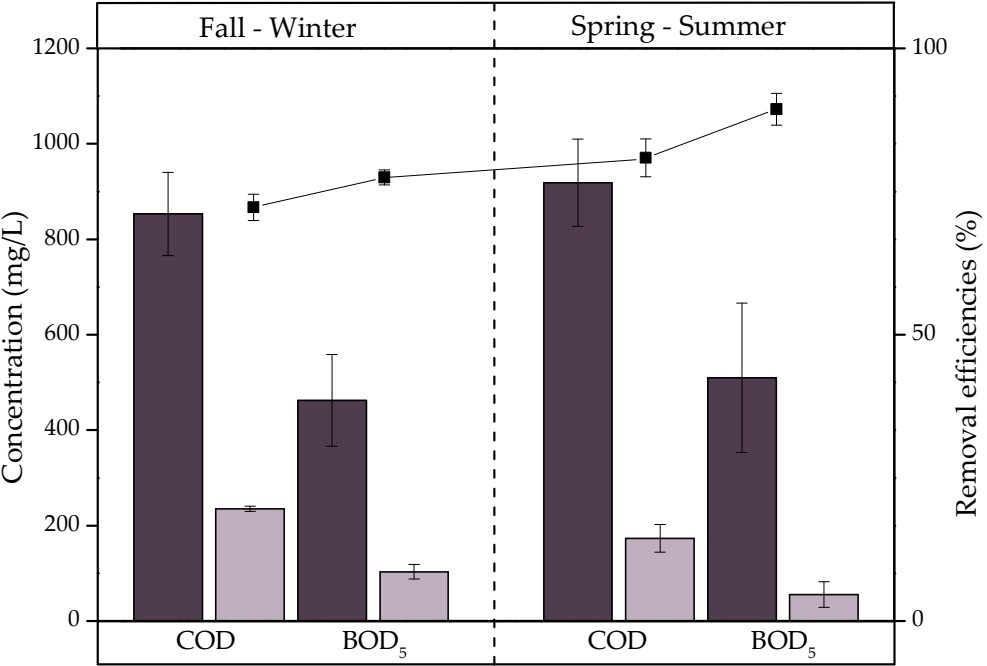

**Figure 4.** Concentrations (mg/L) of COD and $BOD_5$ in IN (■), E2 (■), and removal efficiencies (%) in fall–winter and spring–summer.

### 3.2.2. Nutrients

Figure 5 shows the TN, $NH_4^+$-N, and $NO_3^-$-N concentrations in IN and E2 in the fall–winter and spring–summer periods. The TN removal efficiencies were statistically significant ($p < 0.05$). In the fall–winter period the reduction was 61%, while in spring–summer it was 40%. This difference may be related to the greater growth and metabolic activity that can affect nitrogen transformation processes [42,58]. Meanwhile, during the fall–winter period, there was a slightly significant increase in TN ($p = 0.06$) which could be explained by the seasonal rainfall that carries nitrogenous compounds and changes the carbon–nitrogen ratio [60]. Zhao et al. [35] reported that the optimum carbon–nitrogen ratio is 5:1 to 10:1; changes outside of this range can lead to decreases in removal efficiency. During the fall–winter period, the carbon–nitrogen ratio was $6.0 \pm 0.7$, while during the spring–summer period it was $11 \pm 0.8$. These changes could explain the decrease in TN removal efficiency.

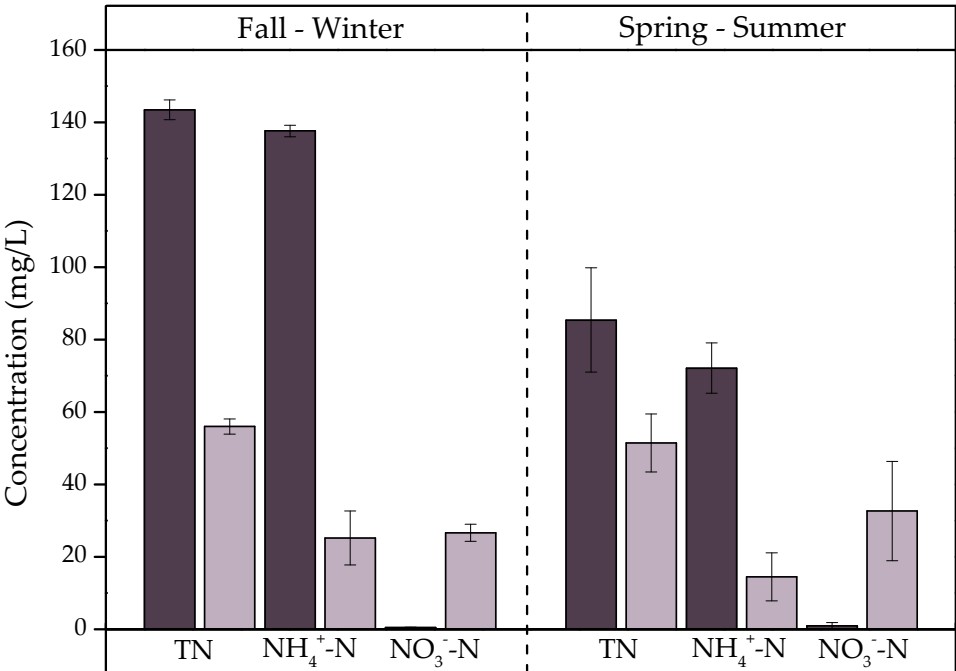

**Figure 5.** Concentrations (mg/L) of total nitrogen (TN), $NH_4^+$-N, and $NO_3^-$-N in IN (■) and E2 (▨) in fall–winter and spring–summer.

### 3.3. Removal of Microbiological Compounds by the Vermifilter

#### 3.3.1. Coliforms

Figure 6 shows the FC and TC concentrations and removal efficiencies obtained in IN, E1, and E2. There were average FC concentrations of $8.9 \pm 8.9$, $6.9 \pm 7.1$, and $6.2 \pm 6.4$ $\log_{10}$(MPN/100 mL), respectively, and average TC concentrations of $8.5 \pm 8.6$, $6.9 \pm 7.3$, and $5.8 \pm 5.8$ $\log_{10}$(MPN/100 mL), respectively.

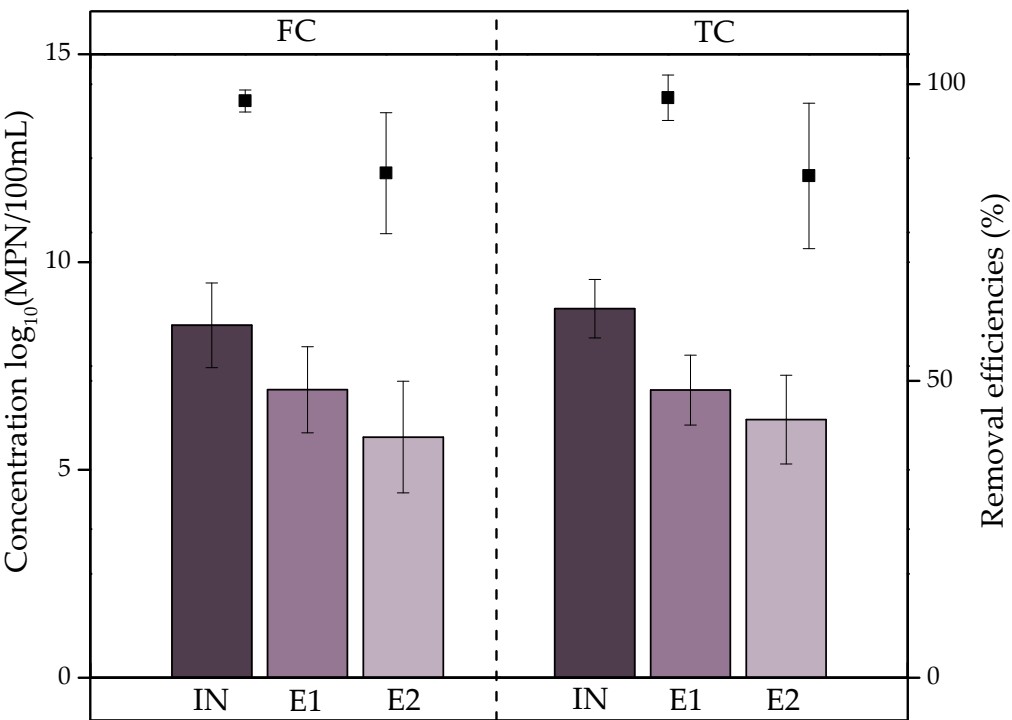

**Figure 6.** Fecal coliform (FC) and total coliform (TC) concentrations ($\log_{10}$(MPN/100 mL)) in IN ( ), E1 ( ), and E2 ( ) and removal efficiencies (%).

The FC and TC removal efficiencies in E1 were on average 97.2% and 97.7%, respectively, which were statistically significant ($p < 0.05$). Arora et al. [23] reported FC and TC removal efficiencies of 99% and 90%, respectively, a result similar to that found by Kumar et al. [22], who reported removal efficiencies of 99.4% and 99.7%, respectively. Arora et al. [9], meanwhile, reported that pathogenic strains such as *Enterobacter*, *E. coli*, and *Pseudomonas* were not detected in the final effluent, showing a considerable removal of pathogens by the VF. These results were achieved with a laboratory-scale VF, that is, under controlled conditions; therefore, they suggest that a full-scale VF could reach these removal levels if its operating and design parameters are optimized.

Coliform removal efficiencies in a VF are associated with filtration, adsorption, and physicochemical conditions inadequate for their survival [16]. Oxygenation and microbial activity increase due to the burrowing activity of the earthworms [9]; therefore, predation and competition can be influential factors in coliform reduction. Wand et al. [61], in studies on predation on *E. coli*, showed removal efficiencies close to 100% due to the activity of flagellated protozoa and *Bdellovibrio*. In addition, earthworms secrete mucus, the viscous and sticky nature of which restricts the movement of foreign microorganisms and aids in their destruction. Das and Paul [62] reported that earthworms ingest and destroy unfavorable microorganisms, decreasing the pathogen count. Finally, the antimicrobial properties of earthworms have already been documented, with 60% of pathogen removal attributed to them [9]. Authors such as Hussain et al. [63] and Chauhau et al. [64], who studied earthworm tissue extracts, body paste, and coelomic fluid, reported that they can act as antimicrobial agents, although they did not exhibit activity against enteric bacteria; therefore, more studies on the behavior of these antimicrobial agents in VFs are needed.

UV disinfection generated FC and TC removal efficiencies of 85.0% and 84.9%, respectively. They were not significant and were lower than those achieved by the VF, which were 12.2% and 13.2%, respectively. González et al. [65] indicate that the presence of TSS (>26.7 mg/L) can influence the efficiency of disinfection treatment. Taking into account that in the present study the TSS concentration was 67.3 mg/L, this could be the reason that the UV treatment was hindered.

The FC and TC removals of the complete WWTP treatment were 99.4% and 98.9%, respectively, which were statistically significant ($p < 0.05$). However, the FC and TC concentrations in E2 were $6.2 \pm 6.4$ and $5.8 \pm 5.8 \log_{10}(\text{MPN}/100 \text{ mL})$, respectively. According to the WHO, the maximum allowable limit in effluents before discharge into rivers is $3 \log_{10}(\text{MPN}/100 \text{ mL})$ [16]. Therefore, the discharge of these effluents into the receiving water body could pose a risk to human and animal health due to the possibility of triggering gastrointestinal diseases produced by *E. coli*, *enterovirus*, *adenovirus*, *salmonella*, and *Crystosporidium* [66]. Tyagi et al. [67], in a study on the removal of pathogens in various WWTPs, determined that the removal of bacteria in sewage is associated with the removal of $BOD_5$ and TSS. Therefore, the optimization of operating parameters and VF design is necessary. Several studies suggest that the removal of organic matter, TSS, and pathogens is enhanced by synchronizing FVs with other treatment systems [17,21,39]. Because of this, further studies are suggested.

### 3.3.2. Antibiotic-Resistant Bacteria

Figure 7 shows the rates of bacterial resistance to AMX, CTX, and CIP in IN, E1, and E2, and the removal efficiencies obtained after the treatments. In IN, the average rates of resistance to AMX, CTX, and CIP were 95.6%, 79.8%, and 71.5%, respectively. This result indicates high resistance rates in rural sewage which have been reported countless times by various authors [7,68,69]. This is attributed mainly to the direct consumption of antibiotics to treat infectious diseases and indirect consumption in the form of contaminated food [70–72]. As a result of this consumption, selective pressure on enteric bacteria can lead to the development of antibiotic resistance [5]. These bacteria are subsequently excreted into sewage together with traces of antibiotics that can continue to select for ARB in the environment [73].

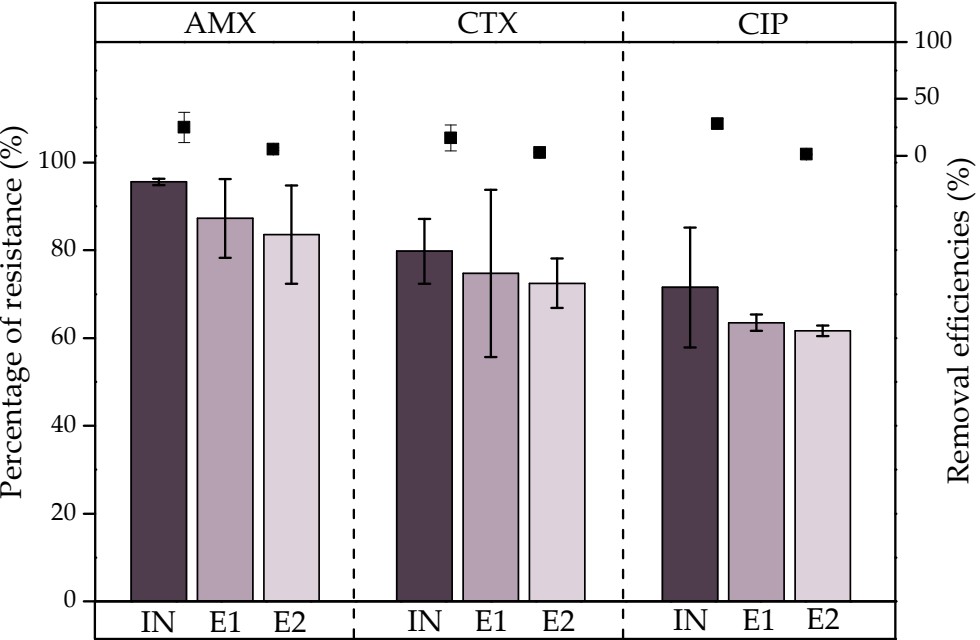

**Figure 7.** Bacterial rates of resistance (%) to amoxicillin (AMX), ceftriaxone (CTX) and ciprofloxacin (CIP) in IN ( ), E1 ( ), and E2 ( ), and ARB removal efficiencies (%).

The resistance rates of E2 for AMX, CTX, and CIP were 87.3%, 74.7%, and 63.5%, respectively, with the VF generating ARB removal efficiencies of 25.0%, 15.7%, and 28.2%, respectively. However, these efficiencies were not significant ($p > 0.05$). Arora et al. [23] reported removal efficiencies of 100% for bacteria resistant to ampicillin, ticarcillin, gentamicin, and chloramphenicol. These analyses were conducted in a pilot-scale VF; therefore, further studies in a full-scale VF are suggested. ARG behavior can vary depending on environmental conditions, parameters, and the operating period [74,75]. Due to biofilm accumulation, ARB and ARG in a pilot-scale system will not have the same results as in a larger system with a longer operating period.

As the ARB counting method is specific to enterobacteria, the detected ARB are considered to belong to the genera *Escherichia*, *Salmonella*, *Enterococcus*, *Shigella*, and *Klebsiella* [6,76]. Considering that the VF significantly removed coliforms and not ARB, this result is associated with the increase in ARB in the system. The presence of selection agents such as antibiotics, heavy metals, and disinfection products raises the risk of an increase in antibiotic resistance, as their ability to select for ARB in the environment has been documented by various authors [77–79]. In addition, the greater oxygenation of the system and the high organic load present could be favoring the transfer of ARG to previously susceptible bacteria [71,80]. The earthworms and microorganisms present form biofilms as a symbiotic association to degrade compounds and other organisms [9,24]. Zhang et al. [81] and Engeman et al. [82] observed a rapid migration of *tet* genes from water columns with animal waste to biofilms. This indicates the capacity of ARG to progressively accumulate in biofilms; therefore, they could act as an ARG reserve.

E2 presented bacterial resistance rates of 83.6%, 72.5%, and 61.7% for AMX, CTX, and CIP, respectively. The ARB removal efficiencies were 5.9%, 2.8%, and 1.4%, respectively. Similarly, it has been documented that disinfection treatments can cause pathogen reactivation, as they have developed DNA repair mechanisms when low doses of UV radiation are applied [65]. In addition, Stange et al. [83] reported average *E. coli* and *E. faecalis* reductions of 5.5 ulog, but insignificant ARG reductions by UV disinfection; therefore, ARG can be transferred to previously susceptible bacteria even after disinfection. Zhuang et al. [84], meanwhile, reported significant removals of *sul*1 and *tet*G; however, they state that the most effective doses of UV radiation are between 10 and 100 times more than those commonly used in WWTP, which would mean high energy consumption. These results indicate that E2 presents a high antibiotic resistance load, both with respect to the assessed ARB and the possible ARG that could be present; therefore, there is a high risk of transmission of antibiotic resistance through the discharge of these effluents into their receiving river. Bueno et al. [85] studied the dissemination of ARG after effluent discharge by a WWTP, finding a significant increase in 17 ARG downstream of a WWTP in Chile. Similarly, Proia et al. [86] observed increases in $bla_{TEM}$, *qnr*S, *tet*O, and *tet*W in the receiving river of two WWTPs in Belgium.

To prevent this spread of antibiotic resistance to bodies of water, authors have investigated the combination of disinfection methods and the use of advanced oxidation processes (AOP). Barancheshme et al. [87] and Zhang et al. [88] report better results using a sequential chlorination/UV disinfection treatment compared to treating each alone. While Zhang et al. [89] and Karaolia et al. [90] determined that AOPs such as $UV/H_2O_2$ and $UV/TiO_2$ can generate ARG reductions of up to 6 ulog, further investigations are recommended both at a full-scale VF and with different sewage characteristics.

## 4. Conclusions

Considering the results of this study, it can be concluded that sewage treatment by means of a full-scale VF operated with 0.5 m$^3$/m$^2$d and 0.6 Kg COD/m$^2$d:

1. Generates statistically significant removals ($p < 0.05$) of COD (77%), BOD$_5$ (84%), TN (53%), and TP (36%). Seasonality is a factor that significantly influenced COD, BOD$_5$, and TN removal. COD and BOD$_5$ are eliminated at 9% and 11% higher rates in spring–summer, respectively, while TN is eliminated at a 21% higher rate in fall–winter. This is because the temperature is influential in the growth and activity of earthworms and microorganisms causing changes in removal efficiencies.

2. The molecular weight distribution indicates that the organic matter (COD and TOC) percentage decreases by an average of 6% in the <1000 Da fraction after the VF, while it increases by an average of 16% in the >10,000 Da fraction; therefore, the VF does not generate considerable changes in the molecular weight distribution of organic matter and NH$_4^+$-N due to processes of degradation, adsorption, and the formation of aggregates with high molecular weights.

3. Coliform removal by the VF was statistically significant ($p < 0.05$) at 99.4% and 98.9% for FC and TC, respectively, although the concentrations in the effluents were 6.2 and 5.8 log$_{10}$(MPN/100 mL), respectively; therefore, the WWTP does not reduce coliforms to safe levels. In addition, ARB removal was not significant ($p > 0.05$); thus, ARB selection and ARG transfer processes are occurring within the system. The effluents discharged into the receiving river could lead to the dissemination of antibiotic resistance in the environment.

4. In consideration of the results obtained, the projections of this research are focused on optimizing full-scale VFs in the elimination of organic matter, nutrients, and pathogens. This includes evaluating different operating and design parameters, determining the efficiency of VFs synchronized to other technologies, and specifying an adequate cost-effective disinfection method for the efficient removal of ARB and ARG.

**Supplementary Materials:** The following supporting information can be downloaded at: https://www.mdpi.com/article/10.3390/su15086842/s1, Table S1, Physicochemical characteristic of the active layer with different earthworm density.

**Author Contributions:** Conceptualization, V.G. and G.V.; methodology, V.G. and G.V.; software, G.G.; validation, G.V., G.G. and N.M.; formal analysis, V.G., N.M. and G.G.; investigation, V.G.; resources, G.V.; data curation, V.G., N.M. and G.G.; writing—original draft preparation, V.G., N.M.; writing—review and editing, V.G., N.M. and G.G.; visualization, G.G.; supervision, G.V.; project administration, G.V.; funding acquisition, G.V. All authors have read and agreed to the published version of the manuscript.

**Funding:** This research was funded by ANID/FONDAP/15130015.

**Institutional Review Board Statement:** Ethical review and approval were not applicable for this study.

**Informed Consent Statement:** Not applicable.

**Data Availability Statement:** The data presented in this study were obtained through laboratory analyses and they were not available in public databases.

**Acknowledgments:** V.G thanks ANID/Scholarship Program/DOCTORADO BECAS CHILE/2021-21211494 for supporting his Ph.D. studies at the University of Concepción.

**Conflicts of Interest:** The authors declare no conflict of interest.

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
