# Peer review of "Performance of a Full-Scale Vermifilter for Sewage Treatment in Removing Organic Matter, Nutrients, and Antibiotic-Resistant Bacteria"

_sustainability, doi:10.3390/su15086842_

Round 1
Reviewer 1 Report
The description of the VF layers seems a little bit confusing (lines 76 thru 84). A drawing indicating the filter layer cross section could be helpful.
Author Response
Dear Reviewer,
The asnwers of the all comments are in the attached file.
Best regards,
Gladys Vidal

Reviewer 2 Report
The authors in this article consider the actual problem of purification of agricultural waters using a vermifilter.
Researchers use modern methods of analysis. the work is framed in compliance with the requirements of the journal.
A large amount of data obtained was processed using statistical methods.
The study addresses the main issue of scaling the vermifilter for treatment of wastewater of agricultural enterprises from organic pollutants. The authors evaluate the effectiveness of this unit.
The topic of the study is relevant, since at present, a large amount of wastewater contaminated with organic substances is formed. The use of environmental treatment methods are effective and inexpensive.
The author's contribution is that the possibility of scaling vermifilter is evaluated. Treatment efficiencies range from 36% for nitrogen compounds to 84% for BOD.
References to literature sources are appropriate.
The conclusions are consistent with the objectives, the conclusion is argued and has evidence.
Author Response

(The authors gave the same response as above.)

Reviewer 3 Report
This experimental research has value and may add to the existing literature, but in its current form the article lacks scientific soundness, and; therefore, some aspects must be improved.
Please check the enclosed file, report the MAJOR comments on a file, answer them and report the amended text.
All the best,

Author Response

(The authors gave the same response as above.)

Reviewer 4 Report
We found the manuscript very pertinent to the ground of this Journal and presents interesting information about a sustainable solution for treating rural sewage. This study evaluates the performance of a full-scale vermifilter aiming to remove/reduce organic matter, nutrients, and antibiotic-resistant bacteria.
The introduction part covers both old and new references concisely (maybe too briefly) and has a perfect integration of the main aspects of the theme.
The cited core references are recent (<5 years) and pertinent to the discussion throughout the manuscript.
This article is well written, with a good organization of the contents and an appropriate experimental design. The monitoring strategy description includes a nice picture with good quality. Concerning the statistical analysis methodology, the authors were careful in verifying the normality of the experimental data. We congratulate the authors on this.
The manuscript is nicely discussed and a brief allusion to the study’s limitations was clearly and succinctly stated in the conclusions.
Specific comments:
Only minor comments will be made relating to the improvement in the fluidity of the discussion around Table 2 [L160] (#Comment 1) and also about the “strange” values of SDs of some variables on the same Table (#Comment 2).
#Comment 1_We suggest the authors kindly include a sign (usually the asterisks) of statistical significance (with the respective level of significance achieved) for each parameter value. The reader will be able to more easily identify the variables with the best results after treatment.
#Comment 2 _SD values are especially high for variables ORP (E2), BOD5 (IN), and TSS (IN), please check if they are correct. If they are correct, do the authors have an explanation for this variability?
Author Response
Dear Reviwer,
The answers of the all comments are in the attached file.
Best regards,
Gladys Vidal

Reviewer 5 Report
The manuscript presents “Performance of a full-scale vermifilter for sewage treatment in removing organic matter, nutrients, and antibiotic-resistant 3 bacteria”. The Authors did great work to assess the organic matter, nutrients, and ARB removal efficiency of a full-scale VF that treats wastewater. In addition, the author also evaluated the performance and the influence of seasonality through molecular weight distribution analysis with respect to organic matter and nutrient removal efficiency.
I am recommending a few minor changes in this manuscript mentioned below.
Line 28: WHO is not used prior as an acronym.
Line 29: “However, in 2020 45% of the wastewater generated in the world was still discharged 29 without safe treatment”. Please modify this.
Line 57: What is meant by CT and CF?
Line 140: MacConckey agar? Provide the manufacturer details of each chemical/material used (Company, City, Country).
Section 2.5: the authors should improve this section. Also mention the purpose of antibiotics either for AST or others.
Author Response
Dear Reviewer,
The answers of the all comments are in the attached file.
Best regards,
Gladys Vidal

Round 2
Reviewer 3 Report
All comments were addressed.